# Effects of Progressive Resistance Training After Hip Fracture: A Systematic Review

**DOI:** 10.3390/jfmk10010054

**Published:** 2025-02-02

**Authors:** Pablo Soro-García, Noelia González-Gálvez

**Affiliations:** Facultad del Deporte, UCAM Universidad Católica de Murcia, 30100 Murcia, Spain; psoro@alu.ucam.edu

**Keywords:** strength, osteoporosis, sarcopenia, physical activity, exercise

## Abstract

Hip fracture presents high morbidity, mortality, and healthcare costs. Some programs have focused on the effect of progressive strength work on post-hip fracture recovery. Therefore, the objective of this systematic review was to understand the effect of a progressive resistance training program on different variables in adults after hip fracture. This review includes randomized controlled trials that apply progressive strength programs in subjects after a hip fracture. The selected databases are PubMed, Cochrane Central Register of Controlled Trials (CENTRAL), and Ebsco. A total of 7 studies were selected after screening. These studies were published between 2005 and 2022. Most of the research included adults over 65 years of age, showing a mean age of 77.80 years. In the majority of cases, the programs are applied between 3 and 12 months post-fracture. The most commonly applied intervention time is 3 months. The intervention time of the programs typically lasts for 3 months and includes 3–4 lower limb exercises involving, mainly hip and knee movements. All the investigations assess functional capacity and nearly all research the strength. It is shown that the intensity of strength work progresses from 60 to 80% of 1RM. Progressive strength training programs in post-hip fracture patients generally show an improvement in functional capacity, strength, balance, walking speed, flexibility, and cardiorespiratory fitness. However, the effects on independence, quality of life, self-reported physical disability, depression, and cognitive ability do not show conclusive results, and there is little research in this regard.

## 1. Introduction

The incidence of hip fracture injuries is so widespread that it poses a significant public health challenge across the globe. This type of injury is associated with considerable morbidity, meaning that it can lead to a range of health complications and a decline in the quality of life for those affected [1,2]. Additionally, hip fractures are linked to high mortality rates, particularly among older adults [3,4].

Furthermore, the financial implications of hip fractures are substantial, as they contribute to increased healthcare costs due to the need for medical treatment, rehabilitation, and long-term care [5,6]. These fractures occur more frequently in older adults, showing a global number of 9.58 million new cases in people over 55 years of age in 2019. This represented an increase of 159.75% since 1990, assuming an incidence of 681.35 per 100,000 inhabitants. The global prevalence was 16.75 million patients over 55 years of age in 2019. Among the causes of hip fracture, there are 3 levels of injury, and its incidence is attributed to falls, road injuries, and exposure to mechanical forces [7].

It is widely recognized that individuals who experience a hip fracture often require a significant amount of time to recover. In most cases, it takes approximately one year for patients to achieve a level of functionality comparable to what they had prior to the fracture [8]. Previous researchers showed that 6 months after hip fracture, only 8% could climb a flight of stairs. Eight months after the fracture, 42% of the sample required a cane or walker, and 54% of the sample reported not walking as well as they could before the fracture. After a 2-year follow-up, patients with hip fracture are 4 times more likely to be confined to their homes and 3 times more likely to be dependent in the basic activities of daily living [9]. Less than half of people living after a hip fracture recover their previous functional status. After a hip fracture, older people have reduced strength, mobility, and balance [10].

As individuals age, there is a progressive decline in both muscle mass and muscle function in a condition commonly referred to as sarcopenia [11]. This natural process of aging is particularly evident in older adults, where the reduction in muscle strength and function contributes to noticeable muscle weakness and increased physical frailty [11]. These individuals have a higher risk of suffering from hip fracture, one of the main causes of physical disability in the older adult population [7,8].

Strength is a strong predictor of hip fractures [12]. In addition, strength influences functional recovery, walking ability, and morbidity after hip fracture [13,14]. The ability to walk is significantly affected by muscle weakness after hip fracture [15]. Muscle weakness reduces the mechanical load on bones, potentially contributing to the development of osteoporosis and increasing the likelihood of hip fractures [12].

Some programs have focused on developing interventions for rehabilitation after hip fracture, incorporating progressive resistance work. These programs display improvements in balance, mobility, strength, or activities of daily living [12]. People who have sustained a hip fracture experience a loss of about 50% of muscle strength on the side of the fractured hip [16,17]. Hip fracture recovery and the recovery of strength in the affected limb are currently [18] challenging. Research has shown promising results in decreasing strength deficits in the affected limb, using progressive resistance work as an important part of the rehabilitation program [19].

On the other hand, sedentary behavior and a low level of physical exercise are also common among patients recovering from hip surgery [20,21]. Postoperative care after hip surgery does not seem to be sufficiently effective [22] in returning people to their previous functional status, as many patients do not achieve their pre-injury functional status.

However, a recent progressive resistance program had no on people after hip fracture, despite incorporating current research and a greater number of variables such as strength, balance, physical disability, depression, independence, quality of life, cardiovascular capacity, functional capacity, etc. Therefore, the objective of this systematic review was to establish the effect of a strength training program on different variables in adults after hip fracture during the rehabilitation period.

## 2. Materials and Methods

### 2.1. Study Design

The methodology for conducting this systematic review adhered strictly to the principles and guidelines outlined in the Transparent Reporting of Systematic Reviews and Meta-Analyses (PRISMA) framework [23]. By following PRISMA, this review was designed to provide a clear and reproducible account of the study’s selection, data extraction, and synthesis processes, thereby enhancing the reliability and validity of the findings.

Furthermore, this research was guided by the comprehensive methodologies detailed in the Cochrane Handbook for Systematic Reviews of Interventions [24].

### 2.2. Eligibility Criteria

The criteria for including articles in the review were clearly defined to ensure the selection of high-quality and relevant studies. These inclusion criteria were as follows: (a) the study design had to be a randomized controlled trial (RCT), a method which is considered the gold standard in clinical research; (b) the study was required to involve the application of progressive resistance intervention programs that specifically target individuals recovering from a hip fracture; (c) the full text of the article had to be accessible to allow for a thorough review and analysis; (d) articles had to be written in either Spanish or English to ensure understanding and accuracy; and (e) only original research articles were considered, with no restrictions on the publication date, allowing for a comprehensive examination of the available evidence.

Conversely, the exclusion criteria were established to refine the selection process further and focus on the intended scope of the review. Articles were excluded if they (a) were related to osteoarthritis, as this was outside the review’s focus; (b) involved strength intervention programs combined with other types of interventions, as these could confound the results; and (c) were categorized as short communications, notes, letters to the editor, review articles, or brief reports, as these formats typically lack the detailed methodology and data required for robust analysis.

### 2.3. Search Strategy

In this context, the information retrieval process was conducted following the standardized guidelines typically applied in reviews of this nature. To achieve a thorough and comprehensive search, the most prominent and reputable databases in the fields of physical activity and health sciences were utilized. The selected databases were PubMed, Cochrane Central Register of Controlled Trials (CENTRAL), and EBSCO, as they are widely recognized for their extensive and high-quality collections of scientific literature in these areas.

The search included all randomized controlled trials (RCTs) published up until 12 November 2024, with no restrictions imposed on the year of publication. This approach ensured that the review encompassed the full breadth of available studies, both recent and historical, to provide a holistic understanding of the topic.

For the search strategy, the primary keyword “hip” was consistently used in combination with other terms related to the intervention and outcomes of interest. These terms included resistance, strength, strengthening, concentric, eccentric, endurance, elastic tube, and pulleys, which were systematically connected using the Boolean AND operator. The full search strategy is shown in Appendix A. This methodological approach facilitated a targeted search that identified studies directly relevant to the research objectives.

Initially, the search yielded a total of 1418 articles. After the removal of duplicate entries, the number was reduced to 898 articles. Subsequently, a screening process was conducted based on the titles of the articles, which further narrowed the selection to 18 articles. The abstracts of these remaining articles were then reviewed, resulting in the exclusion of additional studies and leaving a total of 12 articles. Finally, after a detailed assessment of the full-text documents, the final selection included 7 articles for the review (Figure 1).

This rigorous and systematic search process was designed to ensure that the selected studies were of the highest relevance and quality, thereby providing a strong evidence base for the analysis and conclusions of the review.

### 2.4. Data Collection and Synthesis

Two reviewers (PSG and NGG) independently screened the literature in the selected databases using the search terms, considering the inclusion and exclusion criteria. In case of any discrepancy about the inclusion of a given study, the data extraction or assessment were repeated without looking at the reviewer’s previous information.

### 2.5. Data Extraction

Table 1 show data extraction. This was performed by two reviewers (PSG and NGG) independently. Disagreements were resolved by repeating the data extraction or assessment without looking at the information previously reported by the reviewer. The following data were extracted from each study: author, design, sample, sex, age, setting, time, inclusion criteria, exclusion criteria, intervention, and outcome variables assessed.

### 2.6. Assessment of Risk of Bias and Quality of Evidence Rating

The quality of the selected studies was critically examined following the Cochrane guidelines using ROB2 [25]. The analysis was performed by two authors (PSG and NGG) independently and disagreements were resolved by discussion and further consultation.

## 3. Results

After conducting a thorough screening process, a total of 7 studies were selected for inclusion in the review, as indicated by references [26,27,28,29,30,31,32]. These studies were published over a span of time, specifically between the years 2005 and 2022. On average, each study included 91.63 participants. The experimental groups in these studies typically consisted of around 46 participants on average, and it was noted that there was a higher representation of female participants across the studies.

Most of the research studies included in the review focused on adults who were 65 years of age or older. However, there were two studies that had slightly different criteria, as they included participants who were over 60 years of age, as indicated in references [26,27]. Despite this slight variation, the overall mean age of participants across all the studies was calculated to be 77.80 years.

The rehabilitation programs or interventions included in the studies were primarily implemented between 6 and 12 months after the occurrence of a hip fracture. However, there are two studies that deviated from this typical time window. One of these studies started the intervention just 18 days after the fracture, as stated in reference [28], while the other began its program only 15 days post-fracture, as indicated in reference [32].

The most applied intervention times were 3 months [26,27,28,29,30,31], 6–9 months [30], and 12 months [32]. Session time ranges from 30 to 120 min, with 60 min being the most common. All the intervention programs applied a progressive resistance training program. The intervention programs included exercises for the lower extremities. These mainly involved hip and knee movements, with studies involving 3 exercises [29] or 4 exercises [26,27,30,32]. In addition to strength exercises, some research included functional exercises, such as sitting, getting up from a chair, walking, or climbing stairs [28,31,32]. The protocols included between 6 [29], 8 [26,29], and 15 [28,30] repetitions. The load was stipulated to range from 65% [28] to 70% [30] or 80% [26,27,28,29,30,31,32]. Some research refers to the subjective perception of effort setting it between 12 and 17 [31], progressing the training towards a lower number of repetitions and a higher load every 2 [28], 3 weeks [30], or 4 weeks [29].

Table 2 shows a summary of the variables investigated in the studies included in this systematic review. We detail the number of investigations that found significant differences in each variable after applying each intervention and the number of variables that did not find significant differences.

All the studies included in the review evaluate functional capacity and, with the exception of one specific study [28], all the others report significant improvements in functional outcomes.

The next most studied variable is strength, with 6 investigations including this variable and all of them showing improvements.

Likewise, it is important keep in mind that the two investigations that intervene after 15 [32] and 18 [28] days are the ones that show the least improvement in these variables, both in functional capacity and strength.

The next most researched variables are walking speed, with 5 studies [26,27,28,29,30,31,32], and cardiovascular capacity, with 4 studies [26,27,28,29,30,31]. Regarding walking speed, 3 out of 5 studies find improvements [26,29,32]. In relation to cardiovascular capacity, all of them show improvements [26,28,30,31].

Balance is assessed in 4 studies and flexibility is assessed in 3. Three of the four studies showed improvements for balance [27,30,32] and two of the three studies found improvements for flexibility [29,30]. One study also recorded improvements in self-reported physical function [26]. Independence was assessed by three studies, with no consensus reached among them. One of them showed improvements [30] and two others did not [26,28]. In connection with quality of life, the investigation does not show improvements [30]. Depression was also evaluated by one investigation, showing no improvement [26].

The risk of bias was assessed and the result is presented in Figure 2. Overall, 3 trials show a low risk of bias, while 5 have some concerns about risk of bias. In most cases, the trials show good randomization. However, the biggest problems relate to bias due to missing outcome data and bias in selection for the reporting results.

**Table 1 jfmk-10-00054-t001:** Characteristics of the studies included in the systematic review.

Author	Sample	Age	Intervention Protocol	Time	Exercises, Series, Repetitions, %1RM	Outcome Measures	Assessment Test
Mangione et al., 2005 [26]	EG = 30 CG = 11	Age range = 60–72 EG = 79.8 ± 5.6 CG = 77.8 ± 7.3	Resistance training: bilateral hip extensors; hip abductors and adductors; knee extensors and knee flexors.	Time since fracture = 5–8 months Intervention time = 3 months 2 days/week Session time = 30–40 min	3 exercises, 3 series, 8 repetition, 80%1RM	Physical disability, depression, cognitive capacity, independence, cardiovascular capacity, gait speed, strength.	SF36, geriatric depression scale, mini-exam Folstein, Barthel indx, 6 min walk test, gait speed in GatiMatII, isometric strength of hip extensors, hip abductors, knee extensors and ankle plantar flexors with dynamomenter
Host et al., 2007 [29]	EG = 31 CG = 30 Male = 29 Female = 71	Age range = 79–85	Phase 1 = improve flexibility, balance, coordination, gait speed, strength. Phase 2 = seated work with isokinetic dynamometer on knee extensors, plantar flexors, knee extensors and leg press.	Time since fracture = 6 months Intervention time = 3 months 1–3 days/week Session time = 45–90 min	3 exercise, 1–2 series, 6–8 repetition, 85%1RM	Balance, physical disability, cognitive ability, independence, gait speed, strength.	9-item modified physical performance test
Sylliaas et al., 2011 [30]	EG = 100 (85 female) CG = 50 (40 female)	Age range ≥ 65 EG = 82.1± 6.5 CG = 82.9 ± 5.8	4 exercises: knee flexors, knee extensors, step forward, knee extension.	Time since fracture = 12 months Intervention time = 3 months 2 days in person and 1 in home Session time = 45–60 min	4 exercise, 3 series, 15–8 repetition, 70–80%1RM	Balance, independence, quality of life, cardiovascular capacity, gait speed, functional capacity, flexibility, strength.	BBS test, NEADL daily life, SF12, 6 min walk test, timed up-and-go test, sit-to-stand test, gait speed from 10 m, 1RM knee flexion and extension
Portegijs et al., 2014 [27]	EG = 23 (15 female) CG = 22 (16 female)	Age range = 60–85 EG = 73.8 ± 6.6 CG = 74.1 ± 7.2	Strength exercises: leg press, knee flexors, hip abductor, hip adductor, low plantar flexors. high load and speed.	Time since fracture = 6 months Intervention time = 3 months Session time = 1–1,5 horas	4–5 exercise, maximum repetitions	Balance, gait speed, functional capacity, strength.	BBS test, 20-item soc, gait speed from 10 m, timed up-and-go test, isometric knee extension,
Okoro et al., 2016 [31]	EG = 25 (15 female) CG = 24 (10 female)	Age range = 65–76 EG = 65.15 ± 9.06 CG = 66.33 ± 11.02	Exercises: sitting, standing, blocking steps, climbing stairs, walking, knee extensors.	Time since fracture = 9–12 months Intervention time = 3 months	5 exercise 1 series, 3–10 repetitions	Cardiovascular capacity, functional capacity, strength.	6 min walk test, timed up-and-go test, SCPT test, sit-to-stand test, maximal contraction of the operated leg quadriceps, stair climb performance
Overgaard et al., 2021 [28]	EG = 100 (81 female) CG = 50	Age range = 77–81 EG = 78.3 ± 7.9 CG = 75.7 ± 8.1	Functional exercises: standing, sitting, climbing stairs, walking, prt exercises: leg press and knee extensors.	Time since fracture = 18 days Intervention time = 3 months Session time = 60 min	2 exercise, 3 series, 15–12−10 repetitions, 60–80%RM	Balance, physical disability, independence, quality of life, cardiovascular capacity, gait speed, functional capacity, strength.	SF36, Bathel 20, 6 min walk test, 10 min walk test, timed up-and-go test, maximal voluntary isometric force of knee extension, short physical performance battery
Soukkio et al., 2022 [32]	EG = 61 (50 female) CG = 60 (41 female)	Age range = 65–81 EG = 83 ± 6 CG = 80 ± 7	Multiple-rep ankle weights knee flexors, knee extension, hip extension, hip flexion	Time since fracture = 2 weeks Intervention time = 3 months Session time = 60 min	Multiple repetition, 12–17 RPE	Balance, cognitive capacity, independence, gait speed, functional capacity, strength.	Short physical performance test, mini-mental MMSE, IADL daily life test, handgrip strength

Leygend: CG = control group; EC = exclusion criterion; EG = experimental group;; RPE = rating of perceived exertion.

**Table 2 jfmk-10-00054-t002:** Summary: variables analyzed and results obtained.

Outcomes	Total Number of Studies	Show Significant Improvements	Do Not Show Significant Improvements
Strength	6	6	0
Functional capacity	7	6	1
Gait speed	5	3	2
Cardiovascular disease	4	4	0
Flexibility	3	2	1
Balance	4	3	1
Self-reported F.F./physical disability	1	1	
Depression	1	0	1
Independence	3	1	2
Quality of life	1	0	1

## 4. Discussion

The aim of this systematic review was to determine the effect of a progressive resistance training program on different variables in adults after hip fracture during the rehabilitation period.

The first finding of this review, which is the first systematic review to include studies that apply progressive strength programs during the treatment period, is that the objective to be achieved is to increase the strength of the lower limbs and show significant evidence of improvement. Previous research has indicated that strength is a strong predictor of hip fractures in older adults [12]. It also appears to be a predictor of functional recovery, walking ability, and morbidity after hip fracture [13,14]. Muscle strength is essential for post-hip fracture rehabilitation and low strength values are associated with poor functional recovery [33].

In this regard, the present systematic review also shows positive effects on functional capacity. This is another factor preventing hip fracture. The older adult population experiences natural changes and health problems in the aging process that lead to the loss of functional capacity [33].

Muscle weakness significantly impairs walking ability after a hip fracture. Research indicates that approximately half of patients do not regain their pre-fracture mobility within the first year post-surgery [15]. Additionally, a study published in 2022 found that between 25% and 50% of elderly patients fail to recover their previous levels of independence and walking ability six months after a hip fracture [34]. These findings highlight the critical role of muscle strength in post-fracture rehabilitation and the challenges many patients face in regaining their prior walking capabilities. This variable is of paramount importance in clinical and rehabilitative contexts due to its strong correlation with a wide range of factors that directly impact an individual’s quality of life and independence. Gait speed is a vitally important variable that must be considered because of its relationship with multiple factors, including the functional capacity and mobility of the subject. A decrease in gait speed is associated with increased frailty, dependency, risk of fall, hospitalization, disability, mobility, and morbidity [35].

Muscle weakness causes the low mechanical loading of the bone, which may also promote osteoporosis and hip fracture. Furthermore, low strength is related to poor postural control and balance and increased risk of fall and fall-related injuries, including hip fractures. Furthermore, low levels of strength are related to a frail phenotype [12].

Due to the lack of daily activity, the strengthening of the muscles, and the continuous degeneration of bone quality, the adult population is increasingly weaker and becoming more and more dependent. This is why a preventive protocol that works on the functional capacity of the person is so important. Thanks to this, we can ensure that older adults are autonomous and independent when carrying out the basic activities of daily life. This is directly related to quality of life [34].

Another variable that consistently emerges by consensus in the research is cardiorespiratory fitness following the implementation of the intervention program. While the specific type of physical activity applied in these programs may not traditionally be considered the most effective for producing substantial improvements in cardiorespiratory fitness, evidence indicates that meaningful enhancements in this variable can still be achieved [36]. An elevated cardiorespiratory fitness is associated with improved cardiovascular and muscular health, enhancing balance, coordination, and muscle strength. These improvements reduce the risk of falls, a leading cause of hip fractures, especially in older adults [37].

Enhancing mobility outcomes following a hip fracture is essential for achieving successful recovery [38]. The studies indicate the existence of a consensus regarding the positive impact on improving mobility. Flexibility plays a significant role in influencing an individual’s functional ability and overall quality of life [39].

Another finding of this research is the time that elapsed after the intervention. The results show that practitioners usually wait between 3 and 12 months to apply these programs. Two programs do not require practitioners to wait this long and are applied after 15 [32] and 18 [28] days. Interestingly, these programs show less improvement than those that wait between 3 and 12 months. This result may provide important knowledge and is in line with previous research [40]. Furthermore, the included studies, involving between two and three weekly sessions of progressive resistance training, show improvements in strength and neural capacity after a hip fracture. The notion that this frequency is necessary to achieve improvements in strength and neural capacity is supported by the literature [41,42,43].

On the other hand, it is shown that the intensity of strength work progresses from 60 to 80%, showing consonance with previous research [40]. Previous research has suggested that 60% of 1RM should be the most effective load for beginners and that 80% of 1RM is appropriate for trained people. In this sense, starting strength training at 60% intensity in an untrained population after a hip fracture may be a good, safe, and effective prescription guideline, and progressing to 80% of 1RM to achieve greater improvements in physical capacity has been shown to be effective. In this regard, one review indicates that moderate loads between 60 and 80% of 1RM, performed for 8 to 12 repetitions per set, enhance hypertrophy gains, which are related to strength production [43]. Furthermore, training with high loads, i.e., intensities of 80% 1RM, results in greater neural adaptations compared to training at low loads [44]. This confirms that training at an intensity of 60–80% 1RM is effective for improving muscle strength, with higher loads being necessary to generate neural adaptations.

With respect to the risk of bias, most studies show moderate or low risk, indicating good quality. Future research should improve certain aspects to improve quality. Randomized controlled trials should report on the gaps in data analyzed and provide more information on treatment. Likewise, result analysis should report on differences in changes with respect to the group, and not only within the group, and provide better statistics relating to significance and effect size.

The first limitation of this systematic review is the relatively small number of randomized controlled trials available on this topic. Additionally, there is significant heterogeneity across the studies in terms of the populations included, with variations in factors such as sex and age. The studies also differ greatly in the protocols applied, including the types of progressive strength training exercises used, as well as the small sample sizes observed in some cases. Another important factor to consider is the variation in the time elapsed since the hip fracture occurred and the start of the rehabilitation program, as well as the different durations for which the programs were applied. These variations create challenges in terms of making meaningful comparisons between the different intervention groups. Additionally, limitations include the lack of a registration protocol. Also problematic is the absence of a third reviewer to resolve conflicts in cases of discrepancies during the screening phase. As a result, the generalizability of the findings is limited, highlighting the need for additional, more standardized studies to provide new and more consistent data that support broader conclusions and recommendations.

## 5. Conclusions

Progressive strength training programs in post-hip fracture patients generally show improvements in functional capacity, strength, balance, walking speed, flexibility, and cardiorespiratory fitness. However, the effects on independence, quality of life, self-reported physical disability, depression, and cognitive ability do not show conclusive results, and there is little research in this regard.

## Figures and Tables

**Figure 1 jfmk-10-00054-f001:**
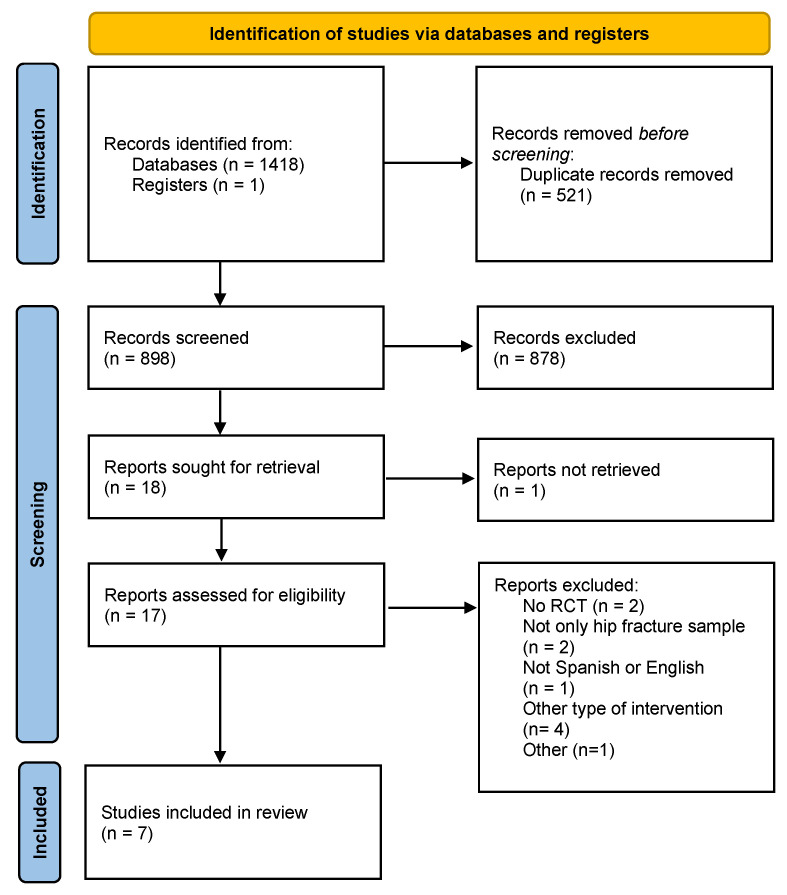
Flow chart of studies searched, selected, and included.

**Figure 2 jfmk-10-00054-f002:**
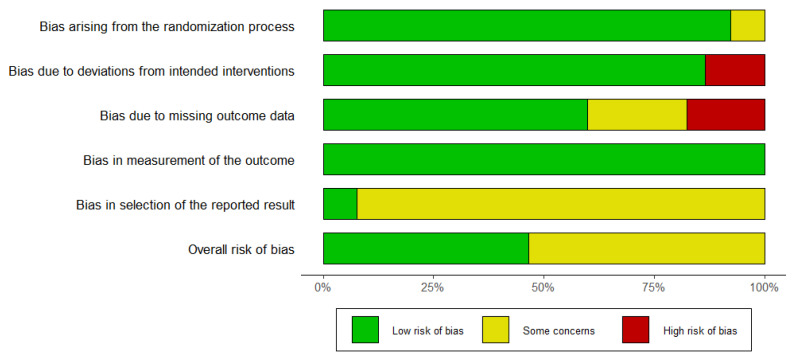
Risk-of-bias assessment.

## Data Availability

The original contributions presented in this study are included in the article. Further inquiries can be directed to the corresponding authors.

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
