# Peer review of "Effects of Progressive Resistance Training After Hip Fracture: A Systematic Review"

_jfmk, 2025, doi:10.3390/jfmk10010054_

Round 1

Reviewer 1 Report

Comments and Suggestions for Authors

Dear authors,

The topic is very important since it can radically affect an individual's life on many levels. However, your submission needs considerable work in order to make a sound case for the study's purpose.  Please see my comments and suggestions below.

Abstract

L17-18: better syntax is needed here.

Introduction -overall comment:

The authors should work more about making a stronger argument for the purpose of their study. Only in the last paragraph do they talk about resistance training, whereas they should start by referring to the loss of strength as a main reason for putting an adult at high risk for falls which will most likely result in a fractured hip. I oppose to the order with which they made their arguments here. I would like to see a better structured presentation of arguments.

L28: it is suggested to write "The incidence of a hip fracture injury is so widespread that..."

L29-31: a couple of references are needed here.

L31-32: please provide relevant references.

L34: remove the dot from "...long-term care."

L43: replace "They" with previous researchers.

L51: unarguably, sarcopenia can have tremendous negative effects for elders. Hip fractures typically happen due to falls as the authors have themselves reported in the 1st paragraph. Falls are the result of impaired muscle strength and neuromuscular coordination due to sarcopenia. Thus, I suggest that instead of writing "quality", the authors write "function".

L61-62: these two phrases need to be placed somewhere else to ensure relevance of context.

L65: "global" evidence? I don't understand how can global evidence come from a resistance training program. It is suggested to remove this word.  

Methods

L148: suggested to write "outcome variables".

Results

L150-153: remove these phrases, they are a repetition of L154-159.

L167-171: remove those lines, they repeat L165-166.

Table 1:

The column defined as "Inclusion and exclusion criteria" should be removed. The selected studies must fullfill these predefined criteria, what is the point of reporting them here since they are quite similar. 

In the introduction, the authors talked about a higher prevalence of hip fracture in females, so I would rather see how each study's sample was distributed per gender. 

Column "protocol intervention group": first of all, rename this column as "intervention protocol" - then, it is highly suggested to provide more data with regards to the training programs used in each study. Provide evidence whenever possible about the training load,  speed of exercise or time of set, as for example for the functional exercises. Also, define number of sets, and time of rest between repetitions and sets. 

Column "variables": it is suggested to rename this column to "outcome measures". It is suggested to insert a new column defined as "Assessment tests" where specific information about how each outcome variable or measure was assessed. The authors write balance, however they are various ways of assessing balance.  

Similarly, what test was used to evaluate gait speed? etc.

Discussion - overall comments:

A considerable part of the discussion is repetition of results. Given that 8 studies were included in this review, the length of the discussion is excessive and should be reduced. There are parts in the discussion where phrases convey the same message, so it is a repetition (for example, L279-281 with L281-283). 

Further, given that the aim of this review was to examine the effect of progressive resistance training programs in older adults after sustaining a hip fracture, the results of those studies with regards to the strength levels of subjects would be the main focus of the discussion. There is no prioritized structure in the the discussion and to my surprise, strength was the last parameter in Table 2. Muscle strength is a proxy measure for other outcome variables of those studies, as for example balance and gait speed. It would be nice if the authors discuss the findings based on a grouping of variables into physiological and/or fitness-related and others, like quality of life, cognitive capacity etc.

Coming back to the results of the included studies with regards to strength, it would be nice to read about training load (for which the authors do talk about) and frequency could have a positive effect not only on sarcopenia but as well as neuromuscular activation for example. This presupposes though that more detailed information about the intervention protocols of the studies be provided. 

L402-403: remove this duplicate reference.

Comments on the Quality of English Language

Please see relevant comments above

Author Response

Dear authors,

The topic is very important since it can radically affect an individual's life on many levels. However, your submission needs considerable work in order to make a sound case for the study's purpose.  Please see my comments and suggestions below.

  • Thank you for your time and your review

Abstract

L17-18: better syntax is needed here.

  • I agree. We changed the sentence L17-18

 Introduction -overall comment:

The authors should work more about making a stronger argument for the purpose of their study. Only in the last paragraph do they talk about resistance training, whereas they should start by referring to the loss of strength as a main reason for putting an adult at high risk for falls which will most likely result in a fractured hip. I oppose to the order with which they made their arguments here. I would like to see a better structured presentation of arguments.

  • Thanks for the comment. The introduction has been reorganized and new arguments about the relationship between strength and hip fracture have been included.

 L28: it is suggested to write "The incidence of a hip fracture injury is so widespread that..."

  • I agree. We changed this sentence L29

L29-31: a couple of references are needed here.        

  • I agree. We included a couple of references L32

L31-32: please provide relevant references.

  • I agree. We included a couple of references L36

L34: remove the dot from "...long-term care."

  • I agree. We remove it L36

L43: replace "They" with previous researchers.

  • I agree. We change it L45

L51: unarguably, sarcopenia can have tremendous negative effects for elders. Hip fractures typically happen due to falls as the authors have themselves reported in the 1st paragraph. Falls are the result of impaired muscle strength and neuromuscular coordination due to sarcopenia. Thus, I suggest that instead of writing "quality", the authors write "function".

  • I agree. We change it L54

L61-62: these two phrases need to be placed somewhere else to ensure relevance of context.

  • I agree. We placed in L 50-51.

L65: "global" evidence? I don't understand how can global evidence come from a resistance training program. It is suggested to remove this word.  

  • We agree. We removed it.

Methods

L148: suggested to write "outcome variables".

  • Thank you. We agree. We are write “outcome variables” L153

Results

L150-153: remove these phrases, they are a repetition of L154-159.

  • Thank you. We agree. We removed.

L167-171: remove those lines, they repeat L165-166.

  • Thank you. We agree. We removed.

Table 1:

The column defined as "Inclusion and exclusion criteria" should be removed. The selected studies must fullfill these predefined criteria, what is the point of reporting them here since they are quite similar. 

In the introduction, the authors talked about a higher prevalence of hip fracture in females, so I would rather see how each study's sample was distributed per gender. 

Column "protocol intervention group": first of all, rename this column as "intervention protocol" - then, it is highly suggested to provide more data with regards to the training programs used in each study. Provide evidence whenever possible about the training load,  speed of exercise or time of set, as for example for the functional exercises. Also, define number of sets, and time of rest between repetitions and sets. 

Column "variables": it is suggested to rename this column to "outcome measures". It is suggested to insert a new column defined as "Assessment tests" where specific information about how each outcome variable or measure was assessed. The authors write balance, however they are various ways of assessing balance.  

Similarly, what test was used to evaluate gait speed? etc.

  • Thank you for the comment. We agree about table 1. We removed the column “inclusion and exclusion criteria”. Information about the distribution by sex has been included in the second column of the table. We have changed the column "protocol intervention group" to "intervention protocol". In addition, we added more columns with more information about the number of exercises, sets, repetitions and training load. No further information has been included because the article does not provide more information. The column "variables" has been renamed to "outcome measures". A new column defined as "Evaluation tests" has been included.

Discussion - overall comments:

A considerable part of the discussion is repetition of results. Given that 8 studies were included in this review, the length of the discussion is excessive and should be reduced. There are parts in the discussion where phrases convey the same message, so it is a repetition (for example, L279-281 with L281-283). 

  • Thank you for your consideration. We agree and we are eliminating the repeated information, and the discussion has been shortened by focusing on the most relevant variables.

Further, given that the aim of this review was to examine the effect of progressive resistance training programs in older adults after sustaining a hip fracture, the results of those studies with regards to the strength levels of subjects would be the main focus of the discussion. There is no prioritized structure in the the discussion and to my surprise, strength was the last parameter in Table 2. Muscle strength is a proxy measure for other outcome variables of those studies, as for example balance and gait speed. It would be nice if the authors discuss the findings based on a grouping of variables into physiological and/or fitness-related and others, like quality of life, cognitive capacity etc.

  • Thank you for your consideration. We agree. The order in Table 2 has been reorganized, and the discussion has been reworked to give greater emphasis to strength, and only those relevant variables have been included in the discussion to give greater emphasis to those important variables.

Coming back to the results of the included studies with regards to strength, it would be nice to read about training load (for which the authors do talk about) and frequency could have a positive effect not only on sarcopenia but as well as neuromuscular activation for example. This presupposes though that more detailed information about the intervention protocols of the studies be provided. 

  • Thank you for your comment. More information on this has been included in the table to be able to understand the discussion and more information have been included in the discussion.

 L402-403: remove this duplicate reference.

  • Thank you. We agree. We have removed it

Reviewer 2 Report

Comments and Suggestions for Authors

-The study needs some adjustments.  The greatest demand would be in some methodological actions and in the description of the results.

-I noticed that the results were presented in a very superficial way. Even though only 8 studies were selected, I recommend exploring the results more in relation to the study's objective.

-The authors cited answers that were unnecessary, such as cardiorespiratory condition, cardiovascular...

-Important adjustments will be needed, which were highlighted in the original document.

Author Response

-The study needs some adjustments.  The greatest demand would be in some methodological actions and in the description of the results.

  • Thank you for your considerations and time invested. They have been taken into account and the document has been improved.

-I noticed that the results were presented in a very superficial way. Even though only 8 studies were selected, I recommend exploring the results more in relation to the study's objective.

  • Thank you for your consideration. We agree. More information has been included in the tables and this has been discussed in the discussion. 

-The authors cited answers that were unnecessary, such as cardiorespiratory condition, cardiovascular...

  • Thank you for your comment. The included studies provide the results of a large number of variables and the present systematic review was intended to provide an overview of the effect of these programs in this population and of what had been previously investigated. For this reason, these variables have been included. However, only those in which the results are found to be significant are mentioned in the discussion in order to prioritize those variables that are most relevant in this type of intervention. In any case, if considered necessary, the necessary variables could be eliminated.

-Important adjustments will be needed, which were highlighted in the original document.

  • Thank you for your comments on the document.
  • The citation is included on line 33.
  • A citation is included on line 54.
  • The wording on line 73 has been changed.
  • More information on the mechanisms is included and further elaborated on lines 59 to 71.
  • Thank you for your consideration of the database, which we will take into account in the future.
  • The redundant parts of the results have been removed.
  • Age was not used as an inclusion criterion. Therefore, information on age in the results section has been included. However, if deemed necessary, it can be omitted.
  • Table 1 provides more information on the intervention programs; this is discussed in the discussion section.
  • The citations are provided in the discussion on the studies that analyse walking speed and cardiovascular capacity.
  • The justification for cardiovascular improvements in the included studies is provided in the discussion.
  • Thank you for your comment on the outcome as a depression. The included studies provide results for a large number of variables and the present systematic review aimed to provide an overview of the effect of these programs in this population and of what has been previously investigated. For this reason, these variables have been included. However, only those in which the results are found to be significant are mentioned in the discussion in order to prioritize those variables that are most relevant in this type of intervention. In any case, if deemed necessary, the necessary variables may be eliminated.
  • The information on the duration of the programs in the discussion has been eliminated based on comments from another reviewer in order to focus the discussion on the salient aspects.
  • The comment in the discussion on cardiovascular capacity has been modified.

Reviewer 3 Report

Comments and Suggestions for Authors

Soro-García and Noelia González-Gálvez performed a systematic review to investigate the effect of a progressive resistance training program on different variables in adults after hip fracture. They identified a total of 8 studies published between 2005 and 2022. The intervention programs included exercises for the lower extremities involving mainly hip and knee movements 3 exercises or 4 exercises. The authors concluded that progressive strength training programs in post-hip fracture patients generally show an improvement in functional capacity, strength, balance, walking speed, flexibility and cardiorespiratory fitness.

The topic is relevant and tackles and important issue in the management of hip fractures. However some limitations are present:

Abstract

- Line 13: please change Cochrane with Cochrane Central Register of Controlled Trials (CENTRAL). See also line 104.

- In the results section, some details about the risk of bias assessment should be provided.

Materials and Methods

- Was the protocol registered in a public database?

- Was age included among the selection criteria?

- Line 137-139: the management of discrepancies during the screening phase is not clear. See also 144-146.

- Risk of bias assessment is completely missing

Discussion

- Line 294. Discuss the statement in light of recent findings in hip surgery.

- Please perform a risk of bias assessment and add an appropriate discussion.

Figure 1: the authors should consider using the official PRISMA flow diagram. In the current version some relevant pieces information are missing.

Please provide the full search strategy used in each database.

Author Response

Soro-García and Noelia González-Gálvez performed a systematic review to investigate the effect of a progressive resistance training program on different variables in adults after hip fracture. They identified a total of 8 studies published between 2005 and 2022. The intervention programs included exercises for the lower extremities involving mainly hip and knee movements 3 exercises or 4 exercises. The authors concluded that progressive strength training programs in post-hip fracture patients generally show an improvement in functional capacity, strength, balance, walking speed, flexibility and cardiorespiratory fitness.

The topic is relevant and tackles and important issue in the management of hip fractures. However some limitations are present:

  • Thank you for your comment and your time.

Abstract

- Line 13: please change Cochrane with Cochrane Central Register of Controlled Trials (CENTRAL). See also line 104.

  • Thank you. We agree. We have change L13 and L106

- In the results section, some details about the risk of bias assessment should be provided.

  • Thanks for the comment. Risk of bias assessment has been included and discussed in the discussion.

Materials and Methods

- Was the protocol registered in a public database?

  • Thank you for the comments. The protocol was not registered in a public database. We will keep this in mind for future occasions.

- Was age included among the selection criteria?

  • Thank you for the comments. No, age was not included among the selection criteria.

- Line 137-139: the management of discrepancies during the screening phase is not clear. See also 144-146.

  • Thank you for the comment. The duplicated information has been eliminate because in both section we inform about the same.

- Risk of bias assessment is completely missing

  • Thanks for the comment. Risk of bias assessment has been included and discussed in the discussion.

Discussion

- Line 294. Discuss the statement in light of recent findings in hip surgery.

  • Gracias por su comentario. La discusión ha sido reelaborada.

- Please perform a risk of bias assessment and add an appropriate discussion.

  • Thanks for the comment. Risk of bias assessment has been included and discussed in the discussion.

Figure 1: the authors should consider using the official PRISMA flow diagram. In the current version some relevant pieces information are missing.

  • Thank you for the comment. Official PRISMA flow diagram has been included.

Please provide the full search strategy used in each database.

  • Thank you for the comment. The full search strategy have been included as a supplementary file 1 and it is indicated in the document.

Round 2

Reviewer 1 Report

Comments and Suggestions for Authors

Dear authors,

Your revised manuscript is considerably improved. 

I have one concern however which, in my opinion, affects the results of your work.

You cited two studies by the same group, of which main author is Sylliaas H.

Both studies have been published in the same journal; the one in 2011 and the other in 2012. The title of both studies is the same, which of course this does not concern you (by the way you need to correct the reference data for the two studies, see note below). However, the methodology and outcome variables present striking similarities -not to mention that the manuscript itself from a fast check I made is also very similar- and the difference basically is the sample size. In the 2011 study, the intervention and control group had 100 and 50 subjects respectively and in the 2012, 48 and 47 subjects in intervention and control groups accordingly. 

I understand the authors' willingness to include as many relevant studies as possible to increase the impact of their work. However, I would personally keep one of the 2 studies as I would be very sceptical about the possibility that the 2012 study was based on subjects whose results had already been published one year earlier.

My recommendation is that the authors include only one study and adjust their manuscript accordingly.

Minor comments

(depending which study remains, these are the correct DOIs:

For refer.nr.30 = Age and Ageing 2011; 40: 221–227 doi: 10.1093/ageing/afq167 

For refer.nr.33 = Age and Ageing 2012; 41: 206–212 doi: 10.1093/ageing/afr164

References nr.12 and 27 are duplicates. Please correct accordingly.

Author Response

Your revised manuscript is considerably improved. 

I have one concern however which, in my opinion, affects the results of your work.

  • Thank you again for your time and dedication

You cited two studies by the same group, of which main author is Sylliaas H.

Both studies have been published in the same journal; the one in 2011 and the other in 2012. The title of both studies is the same, which of course this does not concern you (by the way you need to correct the reference data for the two studies, see note below). However, the methodology and outcome variables present striking similarities -not to mention that the manuscript itself from a fast check I made is also very similar- and the difference basically is the sample size. In the 2011 study, the intervention and control group had 100 and 50 subjects respectively and in the 2012, 48 and 47 subjects in intervention and control groups accordingly. 

I understand the authors' willingness to include as many relevant studies as possible to increase the impact of their work. However, I would personally keep one of the 2 studies as I would be very sceptical about the possibility that the 2012 study was based on subjects whose results had already been published one year earlier.

My recommendation is that the authors include only one study and adjust their manuscript accordingly.

  • Thank you for your comment. We have proceeded to remove the study with citation 33, and have reworked the document accordingly.

Minor comments

(depending which study remains, these are the correct DOIs:

For refer.nr.30 = Age and Ageing 2011; 40: 221–227 doi: 10.1093/ageing/afq167 

For refer.nr.33 = Age and Ageing 2012; 41: 206–212 doi: 10.1093/ageing/afr164

  • Thank you for the comment. We have modified the doi of the study with citation 30.

References nr.12 and 27 are duplicates. Please correct accordingly.

  • Thank you for the comment. We eliminated the reference 12, but 27 we do not find the duplicate.

Reviewer 3 Report

Comments and Suggestions for Authors

The authors significantly improved the manuscript by addressing most of the comments reported. However, some issues still have to be managed:

The lack of protocol registration should be reported in the limitations section.

The limitations should mention the absence of a third reviewer to resolve conflicts in case of discrepancies during the screening phase.

In the limitations section, the authors should describe the specific limitations of their work rather than those of previously published studies. The latter should be mentioned in the discussion as a tool to interpret the findings adequately.

Figure 1. Please revise the reasons for exclusion (i.e., no study design of interest, no population of interest, no intervention of interest). Missing full text should be counted in "reports not retrieved" before the full-text screening.

Author Response

Reviewer 2

The authors significantly improved the manuscript by addressing most of the comments reported. However, some issues still have to be managed:

  • Thank you again for your time 

The lack of protocol registration should be reported in the limitations section.

The limitations should mention the absence of a third reviewer to resolve conflicts in case of discrepancies during the screening phase.

In the limitations section, the authors should describe the specific limitations of their work rather than those of previously published studies. The latter should be mentioned in the discussion as a tool to interpret the findings adequately.

  • Thank you very much for your comments. The discussed limitations have been included in the corresponding section.

Figure 1. Please revise the reasons for exclusion (i.e., no study design of interest, no population of interest, no intervention of interest). Missing full text should be counted in "reports not retrieved" before the full-text screening.

  • Thank you for your feedback. The flowchart has been updated based on your comments.

Round 3

Reviewer 1 Report

Comments and Suggestions for Authors

Dear authors,

Your manuscript's quality has been increased. I have a couple of minor suggestions relating to better wording, at least in my opinion.

L16-17: Consider writing "In their majority, the programs are applied between 3 and 12 months post-fracture".

L17-19: Consider writing: :The intervention time of the programs typically lasts for 3 months and includes 3-4 lower limb exercises involving mainly hip and knee movements".

L229: it is suggested to write "...during the rehabilitation period" or the post-surgery period.

L236: Instead of "research" it is suggested to write "systematic review".

Author Response

Dear authors,

Your manuscript's quality has been increased. I have a couple of minor suggestions relating to better wording, at least in my opinion.

  • Thank you again for your consideration.

L16-17: Consider writing "In their majority, the programs are applied between 3 and 12 months post-fracture".

  • Thank you. The change has been made.

L17-19: Consider writing: :The intervention time of the programs typically lasts for 3 months and includes 3-4 lower limb exercises involving mainly hip and knee movements".

  • Thank you. The change has been made.

L229: it is suggested to write "...during the rehabilitation period" or the post-surgery period.

  • Thank you. The change has been made.

L236: Instead of "research" it is suggested to write "systematic review".

  • Thank you. The change has been made.

Reviewer 3 Report

Comments and Suggestions for Authors

The authors addressed all the issues raised. No further comments.

Author Response

Thank you